# Rapid Automated Screening for SARS-CoV-2 B.1.617 Lineage Variants (Delta/Kappa) through a Versatile Toolset of qPCR-Based SNP Detection

**DOI:** 10.3390/diagnostics11101818

**Published:** 2021-10-01

**Authors:** Dominik Nörz, Moritz Grunwald, Hui Ting Tang, Flaminia Olearo, Thomas Günther, Alexis Robitaille, Nicole Fischer, Adam Grundhoff, Martin Aepfelbacher, Susanne Pfefferle, Marc Lütgehetmann

**Affiliations:** 1Institute of Medical Microbiology, Virology and Hygiene, University Medical Center Hamburg-Eppendorf (UKE), 20246 Hamburg, Germany; d.noerz@uke.de (D.N.); grunwald.moritz@web.de (M.G.); hui-ting.tang@stud.uke.uni-hamburg.de (H.T.T.); f.olearo@uke.de (F.O.); nfischer@uke.de (N.F.); m.aepfelbacher@uke.de (M.A.); s.pfefferle@uke.de (S.P.); 2Virus Genomics, Leibniz Institute for Experimental Virology (HPI), 20251 Hamburg, Germany; thomas.guenther@leibniz-hpi.de (T.G.); alexis.robitaille@leibniz-hpi.de (A.R.); adam.grundhoff@leibniz-hpi.de (A.G.)

**Keywords:** SARS-CoV-2, molecular diagnostics, RT-PCR, variant of concern, B.1.617

## Abstract

Background: The recent emergence of distinct and highly successful SARS-CoV-2 lineages has substantial implications for individual patients and public health measures. While next-generation-sequencing is routinely performed for surveillance purposes, RT-qPCR can be used to rapidly rule-in or rule-out relevant variants, e.g., in outbreak scenarios. The objective of this study was to create an adaptable and comprehensive toolset for multiplexed Spike-gene SNP detection, which was applied to screen for SARS-CoV-2 B.1.617 lineage variants. Methods: We created a broad set of single nucleotide polymorphism (SNP)-assays including del-Y144/145, E484K, E484Q, P681H, P681R, L452R, and V1176F based on a highly specific multi-LNA (locked nucleic acid)-probe design to maximize mismatch discrimination. As proof-of-concept, a multiplex-test was compiled and validated (SCOV2-617VOC-UCT) including SNP-detection for L452R, P681R, E484K, and E484Q to provide rapid screening capabilities for the novel B.1.617 lineages. Results: For the multiplex-test (SCOV2-617VOC-UCT), the analytic lower limit of detection was determined as 182 IU/mL for L452R, 144 IU/mL for P681R, and 79 IU/mL for E484Q. A total of 233 clinical samples were tested with the assay, including various on-target and off-target sequences. All SNPs (179/179 positive) were correctly identified as determined by SARS-CoV-2 whole genome sequencing. Conclusion: The recurrence of SNP locations and flexibility of methodology presented in this study allows for rapid adaptation to current and future variants. Furthermore, the ability to multiplex various SNP-assays into screening panels improves speed and efficiency for variant testing. We show 100% concordance with whole genome sequencing for a B.1.617.2 screening assay on the cobas6800 high-throughput system.

## 1. Introduction

In late 2020 and early 2021, several distinct SARS-CoV-2 lineages emerged and rapidly exceeded the diverse SARS-CoV-2 population of their respective areas of origin. These include current variants of concern (VOC) B.1.1.7 (alpha variant) [1], B.1.351 (beta variant) [2], P.1 (gamma) [3], and B.1.617.2 (delta) variant [4], which has recently replaced the alpha-variant as the predominant lineage worldwide. Classification as VOC often entails increased transmission rates, increased severity of disease, and/or decreased neutralization by pre-existing antibodies against SARS-CoV-2, i.e., antibodies generated by previous infections or vaccinations [5,6]. The requirement to perform whole genome sequencing for lineage assignment of SARS-CoV-2 in clinical samples often conflicts with the need of clinicians and public health institutions to rule out relevant VOCs as fast as possible, e.g., in outbreak scenarios. In this context, surveillance programs can be complemented by rapid RT-qPCR typing tests, which in turn receive continuous updates based on whole-genome sequencing data.

Spike gene mutations such as N501Y, E484K/Q, L452R, and P681H/R have occurred independently in multiple VOCs and VOIs (variants of interest), enabling dynamic adaptation of existing assay designs to detect emerging SARS-CoV-2 variants [7]. RT-qPCR is the gold standard for detection of SARS-CoV-2 in clinical samples [8] and has also been used for SARS-CoV-2 spike gene mutation detection in previous studies, often focusing on the HV69/70 deletion and the N501Y SNP (single nucleotide polymorphism) using various methods such as specific primers, simple probes, and melt curve analyses [9,10,11,12,13,14,15]. Locked nucleic acid (LNA)-Probes represent a well described method for specifically detecting SNPs by RT-PCR [16,17,18].

The aim of the present study was to leverage LNA-probe methodology to create a comprehensive and adaptable design framework for specific SARS-CoV-2 SNP-assays, especially within the spike gene receptor binding domain (RBD) to generate clear yes/no answers for the presence or absence of the respective SNPs. The use of a multi-LNA (locked nucleic acid)-probe concept allows for multiplexing of multiple assays and the option for full automation on a sample-to-result platform to facilitate rapid and efficient screening. Using the described design principles offers the opportunity to adapt multiplex-panels to suit local needs or to screen for novel variants.

## 2. Material and Methods

### 2.1. Assay Design Principles

SNP-assays were designed using PrimerQuest software (IDT, https://eu.idtdna.com/pages/tools/primerquest) (accessed on 29 September 2021). In general, Taqman-probes were adjusted for a melting temperature (Tm) of approximately 63–65 °C (Oligo analyzer, IDT, https://eu.idtdna.com/pages/tools/oligoanalyzer) (accessed on 29 September 2021). at a length of 12–20 bases while containing 5–7 locked nucleic acid (LNA)-bases. LNA-bases feature a 2′O, 4′C-methylene bridge to restrict flexibility of the ribose and generally enhance duplex stability [18]. To achieve high mismatch discrimination, a triplet of LNA-bases is centred on the SNP of interest (Figure 1a) [17]. For each variation of the target sequence, an additional probe (or LNA-blocker oligo) is created and included in the reaction. LNA-Blockers act as a replacement for a competing probe, if only one variation of the respected sequence is to be detected. If melting-temperatures differ widely between probes, additional LNAs are placed (or removed) until melting temperatures are about equal (Figure 1b). For each probe sequence, 2–3 primer-pairs are generated (PrimerQuest) for amplicon sizes of 60–150 bases, with melting temperatures between 60–62 °C. 2′O-methyl RNA bases are placed close to the 3′-end of all primers to inhibit formation of primer-dimers. For assays located within the Spike-gene receptor-binding domain (E484K/ E484Q and L452R), an additional reverse primer is added (“Rev-2”/”RBD-universal rev”, Table 1, Appendix A) to enhance cDNA generation and overall assay performance. If necessary, individual probes were optimized by adding, removing, or changing the position of LNAs to improve signals or reduce off-target activity. Oligos used in this study were custom made by Ella Biotech (Martinsried, Germany), biomers.net GmbH (Ulm, Germany) and IDT (Coralville, IA, USA).

### 2.2. SNP Assays

Based on previous works [19], for detecting del-HV69/70 and N501Y via RT-qPCR, we designed and tested further assays to be able to detect and discriminate all lineages currently holding VOC status (https://www.ecdc.europa.eu/en/covid-19/variants-concern, accessed on 28 June 2021): del-Y144/145, E484K, E484Q, P681H, P681R, L452R, and V1176F. Respective sequences can be retrieved from Appendix A. Different SARS-CoV-2 isolates, either from cell culture supernatant or clinical samples, were used to represent the respective SNPs (I) WT: Hamburg-1 [20], (II) B.1.480: E484K + P681H + delY144/145, (III) B.1.617.1: E484Q + P681R + L452R. All positive material was adjusted to roughly 500,000 RNA-copies/mL (Ref. [21], adjusted to reference material by INSTAND e.V., Düsseldorf, Germany), before 10-fold dilution series were prepared in NA-dilution buffer (Qiagen, Hilden, Germany).

### 2.3. PCR Setup for Proof-of-Concept SNP Experiments

Nucleic acid extraction of clinical samples or cell culture supernatant was carried out using a MagNA-pure 96 instrument (small volume DNA/viral-NA-Kit, Roche, Switzerland). For proof-of-concept experiments, Spike-gene-mutation assays are tested for viability with cobas Utility Channel PCR-chemistry (cobas omni utility channel kit or utility channel optimization kit, Roche, Switzerland). MMX-R2 mastermix was prepared according to instructions by the manufacturer. For proof-of-concept experiments with cobas omni utility channel kit, 3 repeats were tested per dilution step (total volume: 26 µL/reaction). Amplification and detection were carried out on LightCycler480 or cobas z480 analyzer instruments using run-protocols as indicated in Appendix A. Additionally, all assays were tested with a standard qPCR Mastermix (Roche RNA process control kit, total volume: 20 µL), prepared according to manufacturer’s instructions and with a single measurement per dilution step (run protocol, see Appendix A).

### 2.4. Multiplex Assay for Detection of the B.1.617 Lineages: Setup and LoD

In light of the recent emergence of the B.1.617.1/2/3 lineages and classification as VOC [22], a multiplex assay (SCOV2-617VOC-UCT) was compiled from the existing assay list (Appendix A) to create a discriminatory high-throughput screening tool on the cobas6800 system (Table 1). Simultaneous detection of the L452R, E484Q, and P681R SNPs further allows differentiation between the B.1.617.1/3 and B.1.617.2 lineages in a single reaction. For a complete run-protocol see Table 2.

Primers and probes were added to MMX-R2 reagent to form the MMX-R2 mastermix and loaded into cobas omni utility channel cassettes, according to instructions by the manufacturer. Final concentrations for oligos are indicated in Table 1. MMX-R2 already contains the cobas 6800/8800 internal control (IC) assay. The IC-target is automatically added during extraction (packaged RNA-target) to function as a spike-in full process control, similarly to commercial PCR tests for the system by Roche diagnostics. IC assay and target sequences are not disclosed by the manufacturer.

The lower limit of detection for all assays was determined using a clinical sample containing B.1.617.1 lineage (L452R positive, E484Q positive, P681R positive) as reference, adjusted to IU/mL according to WHO standard (NISBSC, UK; https://www.nibsc.org/documents/ifu/20-136.pdf) (accessed on 28 June 2021). A 2-fold dilution series was prepared using pooled negative patient samples (UTM) as matrix (concentrations: 2000, 1000, 500, 250, 125, 61.5, 30.75 IU/mL). Samples were run with a total of 8 repeats per dilution step and 95% probability of detection was calculated by probit analysis (medcalc 20.006, MedCalc Software Ltd., Ostend, Belgium).

For evaluation of exclusivity, a panel (n = 25) of external quality control samples and clinical samples containing various respiratory pathogens was run with the assay, notably containing endemic human coronaviruses, MERS and SARS-CoV (from 2003) (see Appendix A).

### 2.5. Evaluation of Clinical Performance of the SCOV2-617VOC-UCT Multiplex Assay

A total of 233 remnant clinical samples (oropharyngeal or nasopharyngeal swabs in UTM or guanidine thiocyanide) were subjected to testing with the SCOV-617VOC-UCT multiplex assay. Diagnostic samples were from UKE Hamburg, “Labor Dr. Fenner & Collegues” Hamburg and Aesculabor Hamburg and were pre-characterized as positive or negative by various commercial and LDT methods (predominantly the Roche cobas SARS-CoV-2 test). Of these 233 predetermined samples, 39 were SARS CoV2 RNA negative and 194 were SARS-CoV-2 RNA positive by qPCR. Lineages are assigned by whole genome sequencing and next-generation-sequencing (NGS) was performed in collaboration with the HPI (Hamburg, Germany) and Fenner & Collegues Lab. Spike-gene mutations L452R, P681R, E484K, and E484Q were positive in 78, 77, 17, and 7 samples, respectively. Consequently, there were a total of 179 detectable spike-gene SNPs within the sample-set. Ninety-six samples were positive for SARS-CoV-2 but did not feature any of the tested mutations. See Table 4 for a complete list of lineages included in each category. 

This work was conducted in accordance with §12 of the Hamburg hospital law (§12 HmbKHG). The use of anonymized samples was approved by the ethics committee, Freie und Hansestadt Hamburg, PV5626.

## 3. Results

### 3.1. RT-PCR Assays with Competitive LNA-Probes Are Highly Specific for Individual Spike-Gene SNPs

As part of this study, the following RT-PCR assays were evaluated for technical viability: E484K/Q, P681H/R, L452R, V1176F, del-Y144/145. With one exception, all probes generated only very little, or no unspecific signals for off-target sequences, as indicated in Appendix A. The only assay showing significant off-target activity was E484-WT with the E484K sequence, which was eliminated by setting an end-point fluorescence cut-off.

The addition of blocker-oligos for positions not covered by specific probes helped to further suppress off-target signals (for “WT”-variants or other known sequence variants in the SNP region) and allow lower relative-fluorescence-increase (RFI)-thresholds (data not shown). As indicated above, blocker-oligos represent a replacement for competing probes (e.g., when used for genotyping human DNA), which would otherwise help to reduce digestion of off-target probes.

Assays were primarily developed for, and tested with, cobas omni utility channel chemistry. To demonstrate broader applicability, all assays were also tested with Roche one-step RT-PCR mastermix (RNA Process control kit). L452R was the only assay to show a severe delay in CT and would require further optimization.

### 3.2. Analytical Performance of the SCOV-617VOC-UCT on the Cobas6800 System

The LoDs for individual targets were determined as follows: 182 IU/mL for L452R (95%CI: 1472–120 IU/mL), 144 IU/mL for P681R (95%CI: 197–91 IU/mL), and 79 IU/mL for E484Q (95%CI: 18.9–139 IU/mL) (Table 3). This implies that LoDs of individual targets are close enough to each other to avoid miss-calls due to individual assays failing at low concentrations of viral RNA. Furthermore, individual assay LoDs indicate at which concentrations of viral RNA negative SNP-assay results can be considered reliable. Exclusivity testing of 25 respiratory pathogens yielded no false positive results.

### 3.3. Clinical Evaluation of the SCOV2-617VOC-UCT

A total of 233 remnant samples were subjected to testing with the SCOV2-617VOC-UCT on a cobas6800 instrument. In 39 negative clinical samples, no false positives occurred. The positive sample set (n = 194) was selected to contain various off-target and on-target lineages, including 7 available B.1.617.1 (kappa) and 67 B.1.617.2 (delta) samples (see Table 4). All SNPs were detected correctly by the assay (179/179 SNPs) (Table 5).

## 4. Discussion and Conclusions

Interest in identification and tracking of novel SARS-CoV-2 variants remains high because of potential implications for treatment- and vaccine effectiveness, as well as public health measures [1,5,6]. This is exemplified by the recent spread of the delta variant in Scotland and England since May 2021. While mutations are expected to accumulate over time as the viral population diversifies, specific positions within the viral Spike-gene have been found to carry the same, or similar mutations in a number of independent, highly successful lineages. These include SNPs and deletions in the N-terminal domain (NTD) [23,24], the receptor-binding domain (RBD) [7] and furin cleavage site [25]. For example, the recently emerged B.1.617 (first detected in India) lineages feature E484Q (only B.1.617.1/3) and P681R SNPs instead of the more common E484K and P681H [22]. The predictability of SNP locations and otherwise high relative stability of the Coronavirus-genome [26,27] creates the opportunity for rapid pre-typing of SARS-CoV-2 in clinical samples by RT-qPCR. 

PCR-based SARS-CoV-2 typing is currently gaining traction as methodology is being published and commercial providers start to offer ready-made tests [13]. Interestingly, interactions of NTD deletions with a popular commercial PCR assay (TaqPath SARS-CoV-2 Assay, Thermo Fischer, Waltham, MA, USA) first directed attention to the expanding B.1.1.7 lineage in Great Britain, thereby hinting at the potential utility of specific qPCR tests for variant prediction [28,29]. The recent surge in publications regarding PCR-typing demonstrates the wide spectrum of potential methods and their varying degrees of suitability for practical use in diagnostics [9,10,11,12,14]. 

The methodology presented in this study provides some key advantages, including direct detection of relevant SNPs and highly specific, dichotomous results, as well as the capability to combine multiple SNP targets for multiplex panels. Implementation on a fully-automated high-throughput sample-to-result platform like the cobas6800 further improves speed and scalability (sample-to-result time of approx. 3.5 h and throughput of 386 samples per 8 h [30]). Depending on local needs, users may be able to use existing SNP-assays to assemble customized multiplex-panels for circulating lineages, or use the design principles outlined above to implement additional assays when new Spike gene SNPs become relevant in the future. Applicability with different RT-PCR chemistry may require additional optimization and restrict multiplexing. In general, 2′O-methyl-RNA bases are tolerated to a varying degree by different polymerases and the removal of these modifications may improve performance for some conventional qPCR-mastermix-products. We have demonstrated the viability of the presented assay set with “RNA process control kit” (Roche, Switzerland), but validation has to be performed separately for each RT-PCR mastermix before use.

Furthermore, it has to be noted that typing results of RT-qPCR based methods represent a preliminary call and definitive assignment of lineages should only be made based on whole genome sequencing. Likewise, primer and probe sequences should regularly by checked for mismatches with contemporary SARS-CoV-2 lineages as additional mutations may occur over time.

In conclusion, we have demonstrated the viability and performance of a competitive-probe RT-PCR design concept for detection of SARS-CoV-2 Spike-gene mutations. As a practical application of the experimental assays provided in this study, we compiled and provided clinical validation for a high-throughput multiplex-assay to specifically screen for B.1.617 lineage variants (delta/kappa) by detecting the L452R, E484Q and P681R single nucleotide polymorphisms, implemented on a fully automated sample-to-result platform. The assay correctly detected all respective SNPs in a clinical sample set, including 67 B.1.617.2 and seven B.1.617.1 samples. The versatility of this methodology enables diagnostic labs to rapidly react and implement screening tests for emerging SARS-CoV-2 variants.

## Figures and Tables

**Figure 1 diagnostics-11-01818-f001:**
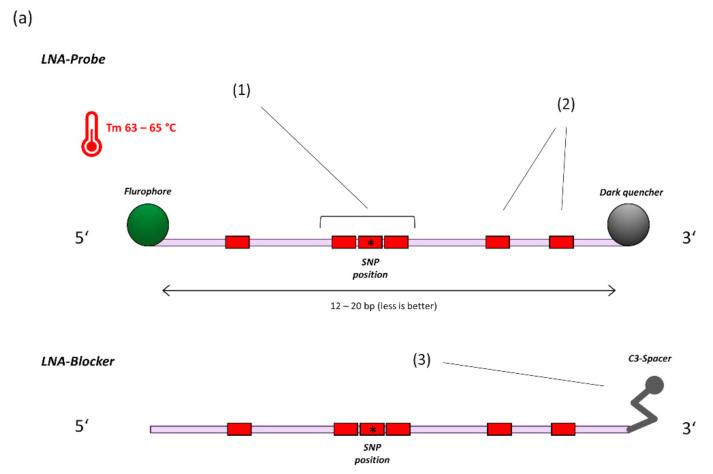
(**a**) General principle of probe design for SNP detection (14). 1. A triplet of LNA bases is centred on the SNP location. 2. Additional LNA-bases are placed based on hairpin and homodimer prediction to achieve a melting temperature of 63–65 °C at a length of 12–20 bases. 3. For target sequences that are not covered by specific probes, a blocker-oligo is created using the same principles but replacing the Fluorophore-Quencher modifications with a 3′ C3-Spacer to prevent elongation. (**b**) 4. Multiple SNP specific probes compete for binding at the target sequence. A perfectly matching probe will bind preferentially and generate signal. 5. Example amplification curves on the LightCycler480 Instrument (E484K/Q assay). With sufficient optimization, off-target probes will generate virtually no signal, thus providing highly specific dichotomous readouts.

**Table 1 diagnostics-11-01818-t001:** Primers and probes were custom made by Ella Biotech GmbH (Martinsried, Germany), biomers.net GmbH (Ulm, Germany) and IDT (Coralville, IA, USA). Indicated final concentrations refer to concentrations within the final reaction mix. OMe-N, 2′O-methyl-RNA base. (+N), Locked nucleic acid (LNA)-base.

Oligo Type	Oligo Name	Sequence 5′-3′	Final Concentration [nM]
Primers	L452R fwd	GAT T(+C)T AAG GTT GGT GG(2OMe-U) AAT	400
	L452R rev	TTT CAG TTG AAA TAT CT(+C) TC(2OMe-U) C	400
	E484K/Q fwd	CTA TCA GGC CGG TAG (OMe-C)A	400
	E484K/Q rev	GTT GGA AAC CAT ATG ATT GTA AA(OMe-G) G	400
	P681R fwd	TGC AGG TAT ATG CGC TAG T(OMe-U)A	400
	P681R rev	GTG ACA TAG TGT AGG CAA TGA (OMe-U)G	400
	RBD-universal rev	AGT TGC TGG TGC ATG TA(OMe-G) AA	400
Probes	L452R-probe	Atto425- T(+T)A C(+C)(+G) (+G)TA TAG ATT (+G)TT TA(+G) -BHQ1	75
	E484K-probe	YakYellow- AT(+G) GTG T(+T)(+A) (+A)AG (+G)TT -BHQ1	75
	E484Q-probe	Atto620- AT(+G) GTG T(+T)(+C) (+A)AG (+G)TT -BHQ2	75
	P681R-probe	FAM- A(+T)T CT(+C) (+G)(+T)C GGC G -BHQ1	75
Blockers	L452WT blocker	T(+T)A C(+C)(+T) (+G)TA TAG ATT (+G)TT TA(+G) -C3-Spacer	75
	E484WT blocker	AT(+G) GTG T(+T)(+G) (+A)AG (+G)TT -C3-Spacer	75
	P681WT blocker	TAA (+T)TC T(+C)(+C) (+T)CG GCG -C3-Spacer	75
	P681H blocker	TAA (+T)TC T(+C)(+A) (+T)CG G(+C)G -C3-Spacer	75

**Table 2 diagnostics-11-01818-t002:** Run protocol for the SCOV2-617VOC-UCT assay, configured with cobas omni Utility Channel software (Roche Diagnostics, Basel, Switzerland). Material type was Swab, 400 µL sample input volume. RFI (relative fluorescence increase) is used as threshold for automatic result calls.

Software Settings					
Sample Type	Swab (400 µL)				
Channels	1: L452R	2: P681R	3: E484K	4: E484Q	5: IC
RFI	1.3	2.8	1.3	2	2
PCR cycling conditions					
	UNG incubation	Pre-PCR step	1st measurement	2nd measurement	Cooling
No. of cycles	Predefined	1	5	45	Predefined
No. of steps		3	2	2	
Temperature		55 °C; 60 °C; 65 °C	95 °C; 55 °C	91 °C; 58 °C	
Hold time		120 s; 360 s; 240 s	5 s; 30 s	5 s; 25 s	
Data acquisition		None	End of each cycle	End of each cycle	

**Table 3 diagnostics-11-01818-t003:** A clinical sample containing B.1.617.1 lineage was normalized to WHO standard. A total of 8 repeats were tested per dilution step. LoD was determined by logistic regression (probit analysis).

SARS-CoV-2 B.1.617.1 Lineage (Kappa)				
Step	IU/mL	L452R: pos/rep	P681R: pos/rep	E484Q: pos/rep
1	2000.00	8/8	8/8	8/8
2	1000.00	8/8	8/8	8/8
3	500.00	8/8	8/8	8/8
4	250.00	8/8	8/8	8/8
5	125.00	7/8	7/8	8/8
6	62.50	3/8	7/8	7/8
7	31.25	5/8	4/8	7/8

**Table 4 diagnostics-11-01818-t004:** SARS-CoV-2 lineages included in the clinical sample set. Lineages were assigned based on whole genome sequencing, as part of the Hamburg Genome Surveillance Project (https://www.hpi-hamburg.de/en/, accessed on 28 June 2021). The total number of SARS-CoV-2 RNA positive samples (any lineage) was 194.

Clinical Sample Set—Included Lineages.		
SNP Set	Lineage	Number
L452R	B.1.617.1	7
	B.1.617.2	67
	C.36	3
	C.16	1
P681R	B.1.617.1	7
	B.1.617.2	67
	A.23.1	3
E484K	B.1.351	7
	B.1.1.28 P.1	4
	B.1.1.318	3
	B.1.525	2
	B.1.1.523	1
E484Q	B.1.617.1	7
Negative for tested SNPs	B.1.1.7	69
	B.1.177	17
	B.1.177.86	2
	B.1.177.81	1
	B.1.221	1
	B.1.1.29	1
	B.1.243	1
	B.1.1.244	1
	B.1.160	1
	B.1.258	1

**Table 5 diagnostics-11-01818-t005:** Clinical samples (UTM or guanidine thiocyanide) were subjected to testing on the multiplex-screening-assay. Results are analyzed for each target SNP individually. A complete list of the involved SARS-CoV-2 lineages can be found in Table 4. The total number of SARS-CoV-2 RNA positive samples was 194.

Target	Result	SNP Positive	SNP Negative	Agreement
L452R	Positive	78	0	100%
	Negative	0	116	100%
P681R	Positive	77	0	100%
	Negative	0	117	100%
E484K	Positive	17	0	100%
	Negative	0	177	100%
E484Q	Positive	7	0	100%
	Negative	0	187	100%

## Data Availability

Data is available upon request.

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
