# Peer review of "Rapid Automated Screening for SARS-CoV-2 B.1.617 Lineage Variants (Delta/Kappa) through a Versatile Toolset of qPCR-Based SNP Detection"

_diagnostics, 2021, doi:10.3390/diagnostics11101818_

Round 1

Reviewer 1 Report

General Comments
The authors describe a qPCR protocol that can better discriminate between SARS-CoV-2 B.1.617 variants of concern by using unlabeled blocker oligos together with Taqman probes containing locked nucleic acids.  It is further designed to recognize and distinguish multiple different SNP mutations in the variants, all at one time. They note that the protocol can quickly be optimized for new variants.

Mechanistically, fluorescent Taqman probes containing a quencher will fluoresce when bound to the appropriate mutation containing sequence during amplification of the relevant region of COVID spike protein RNA during qPCR. The probes for each mutation are chosen to have fluorophores of different wavelengths from each other but, along with their primers, all are designed to bind at around the same temperature.  This allows probes and primers for different SNPs to be PCRed and detected together, saving on patient sample and greatly reducing the time and effort it takes to determine which variant a patient has.

Probe binding is stabilized in the SNP region and some other places by replacing the probe’s nucleic acids in those areas with locked nucleic acids. Unlabeled blocker probes are also added for cleaner discrimination in binding of the appropriate probes between SNPs and the original sequence.  The 3’ ends of the primers contain 2’O-methyl RNA bases to help prevent other unwanted amplification products.

There are a few points that need to be taken into consideration. However, overall, the study is sound, and the paper is mostly well written and timely.

Specific Comments
Major
There is absolutely no description of the internal control.  What is it, sequence, concentration, more detail on how used, etc.?

Line 69- “For each variation of the target sequence” If you mean one blocker oligo with the original non-SNP sequence, this is not clear.  See below comment for Line 168-170.
Line 139-142- At what IU/concentration? i.e., Since results were negative, how do you know there was enough virus used for the assay to work, if the virus were to contain a sequence recognized by one of your probes?
Line 168-170- It needs to be made much clearer how the blocker oligos work. “unspecific signals“ implies that the blocker is preventing the fluorescent probe from binding in regions other than SNP containing region.  In fact, it is mainly preventing the fluorescent probe from binding the region where it’s specific SNP would be but only when the SNP is not there and instead, the pre-mutated sequence is there, that perfectly matches the blocker oligo sequence.

Minor
Line 106- The right side of Table 1 is cut off, one can't see the entire sequence of primers and probes, especially which quenchers were used.  Move the Sequence information to the left so that the entire sequence can be viewed.

Table 2- A lot of the information in this Table 2 and supplementary Table 2 is redundant.  Switch Supplementary Table 2 into the text and make this table become Supplementary 2. It is easier to understand.  Furthermore, in the Channels row, why are the title values for 1 bolded and italicized, 2 only bolded, 3 through 5 not bolded, and 5 italicized?  Please make them all the same font or explain why they are different.

Line 254-256- It would be helpful to put the abbreviation definitions at the beginning of the paper where they can be quickly found, rather than between the acknowledgments and the authors’ contribution.  Some additional abbreviations also need to be defined: 
Line 19- what does UCT stand for?
Line 122-124- UNG abbreviation (uracil-DNA N-glycosylase) should be defined either in Table 2 or with the other abbreviations

A few grammatical errors and wrong word choices were missed and need to be corrected:
Line 35- “…as the dominant…”
Line 71- “…to equilibrate melting temperatures of all probes…”  Wrong usage of the word "equilibrate", should be something like "until the melting temperatures of all probes are about the same”.
Line 90- “Exemplary amplification curves…”  Wrong usage of the word “Exemplary”, should be “Example amplification curves”.
Line 188-189- “SCOV2-617-UCT” should be “SCOV2-617-VOC-UCT”

Reviewer 2 Report

This manuscript describes the development of a method for the detection of multiplexed spike-gene of SARS-CoV-2. The method may be used for improving the speed and efficiency of SARS-CoV-2 variant testing. The work was conducted in accordance with the Hamburg hospital law and the use of anonymized samples was approved by the ethics committee. The manuscript is suitable for publication in Diagnostics after minor revision. Below is a list of some revisions that should be addressed to improve the quality of the manuscript before it is accepted.

  1. General: This is a pleasingly well-written and interesting manuscript.
  2. General: The introduction part does not provide enough information about the current stage of SARS-CoV-2 diagnostics/detection. Thus additional information should be added here. There is significant prior literature work on SARS-CoV-2 detection/diagnostics and should be cited e.g., 10.1021/acsanm.0c01978.
  3. Lines 122-125: Table 2 is divided into two parts on two separate pages. All table 2 and its caption should be presented within 1 page.
  4. Lines 165-166: The authors should explain why "The only assay showing significant off-target activity was E484-WT 165 with the E484K sequence" and provide literature citations that support the reason(s) if possible.
  5. Line 182: The is a formatting mistake here ("SARS-CoV-2 B.1.617.1 lineage ").

     6. Addition information detailing the time for each detection/diagnostic should be provided (adding to Tables 3, 4, 5, and/or mention in the text).

Reviewer 3 Report

The manuscript describes a very interesting method to rapidly detect variants of the B.1.617 lineage.

This method can be applied to all possible variants, but it is not demonstrated in the experiments presented. So I think they should focus on the delta variants.

For example, it is true that they created a toolkit for multiplexed detection of SNPs from the Spike gene, but they only showed data for B.1.617 (although other variants can be classified). Therefore, on line 15 of the abstract it will be more appropriate to end the sentence with something similar to “discriminate lineage variants B.1.617”.

This idea can be reflected in the discussion.

 I have some comments and/or doubts. 

Abstract

I mentioned above, the end of the sentence on line 15.I think line 22-23 is part of Methods (Material), not results.I don't really understand the number of samples studied: 206 or 233. And how they were processed or valid: 194 or 179Of these 179, correctly identified, they were only 74, right?Wouldn't it be better to distinguish between delta and non-delta variants?

Introduction

In this field, one of the first manuscript reported was
 Sandoval Torrientes M, Castelló Abietar C, Boga Riveiro J, Álvarez-Argüelles ME, Rojo-Alba S, Abreu Salinas F, Costales González I, Pérez Martínez Z, Martín Rodríguez G, Gómez de Oña J, Coto García E, Melón García S. A novel single nucleotide polymorphism assay for the detection of N501Y SARS-CoV-2 variants. .J Virol Methods. 2021 Aug;294:114143. doi: 10.1016/j.jviromet.2021.114143. Epub 2021 Mar 24.PMID: 33774075 

Methods

In point 2.5, why is 206 samples instead 233?.In line 154:  96 or 95 (table 5, negative for tested SNPs) samples were positives. 

Results

In line 177 from point 3.2, data 1472 is correct?Point 3.1 wouldn´t it be methods?In this way, Lines 187-189 from point 3.3 are a part of material.I understand that the authors studiing 233 samples. From them 194 were positive. In 179 SNP methods and NGS were performed. Is it true?I think that table 4 and 5 should be merged. And I suggest the authors make a division between B.1.617 and others variants.

Discussion

I suggest moving lines 240-241 to line 247, before the last sentence. And change identified by "classified" or similar, because it only identifies the strains with sequencing methods
